# Reconfiguration of interfacial energy band structure for high-performance inverted structure perovskite solar cells

Moyao Zhang[1], Qi Chen [2]*, Rongming Xue[1], Yu Zhan[1], Cheng Wang[2], Junqi Lai[2], Jin Yang[2], Hongzhen Lin[2], Jianlin Yao[1], Yaowen Li[1]*, Liwei Chen [2,3] & Yongfang Li[1,4]

Charged defects at the surface of the organic–inorganic perovskite active layer are detrimental to solar cells due to exacerbated charge carrier recombination. Here we show that charged surface defects can be benign after passivation and further exploited for reconfiguration of interfacial energy band structure. Based on the electrostatic interaction between oppositely charged ions, Lewis-acid-featured fullerene skeleton after iodide ionization (PCBB-3N-3I) not only efficiently passivates positively charged surface defects but also assembles on top of the perovskite active layer with preferred orientation. Consequently, PCBB-3N-3I with a strong molecular electric dipole forms a dipole interlayer to reconfigure interfacial energy band structure, leading to enhanced built-in potential and charge collection. As a result, inverted structure planar heterojunction perovskite solar cells exhibit the promising power conversion efficiency of 21.1% and robust ambient stability. This work opens up a new window to boost perovskite solar cells via rational exploitation of charged defects beyond passivation.

[1] Laboratory of Advanced Optoelectronic Materials, College of Chemistry, Chemical Engineering and Materials Science, Soochow University, Suzhou 215123, China. [2] i-Lab, CAS Center for Excellence in Nanoscience, Suzhou Institute of Nano-Tech and Nano-Bionics, Chinese Academy of Sciences, Suzhou 215123, China. [3] In-situ Center for Physical Sciences, School of Chemistry and Chemical Engineering, Shanghai Jiaotong University, Shanghai 200240, China. [4] CAS Research/Education Center for Excellence in Molecular Sciences, Institute of Chemistry, Chinese Academy of Sciences, Beijing 100190, China.
*email: qchen2011@sinano.ac.cn; ywli@suda.edu.cn

Organic–inorganic perovskite solar cells (pero-SCs), consisting of a polycrystalline perovskite active layer sandwiched between electrodes, have drawn great attention because of the rapid increase in power conversion efficiency (PCE)[1–8]. Owing to the low defect formation energy of perovskite materials, high density of defects are likely to form in ionic perovskite, including positively charged under-coordinated Pb$^{2+}$, negatively charged Pb-I antisite defects (PbI$_3^-$), and under-coordinated halide ions[9–13]. Most of these defects are benign other than those at the surface of the perovskite, which act as charge traps to exacerbate trap-assisted recombination[14–16], trigger ion migration[17,18], and degrade perovskite[19]. Even worse, surface defects give rise to gap states, which tend to pin the Fermi level of the sequent contacting layer via charge transfer[20]. The resultant trapped charges at the surface can produce localized energy level offsets, leading to undesirable energy band structure and lower built-in potential ($V_{bi}$)[21,22]. Therefore, surface defects should be taken into account for enhancing the PCE and stability of pero-SCs[23–25].

Taking the charged nature of perovskite surface defects into consideration, organic compounds containing a Lewis base (e.g., thiophene, pyridine, phosphate, and halide)[14–16,26–29] or Lewis acid (e.g., fullerene)[12,17] moieties have been demonstrated to effectively passivate the positively or negatively charged defects via coordination or electrostatic interactions. Despite the mitigated hysteresis and enhanced long-term stability after surface passivation, the PCE of these devices still needs much improvement due to undesired deterioration in the interfacial energy band structure[20]. For example, organic passivation agents with low mobility are likely to introduce serious energy disorder on the perovskite surface, giving a wider distribution of band tail states, which lower the open-circuit voltage ($V_{oc}$) due to the reduced upper limit of quasi-Fermi level splitting[20]. Recently, pero-SCs with wide-bandgap perovskites stacked onto a narrow-bandgap perovskite active layer exhibited superior PCEs, in which wide-bandgap perovskites not only passivated surface defects of the perovskite active layer but also tuned the interfacial energy band structure to selectively conduct one type of charges while block the other type of charges[30–32]. However, the introduction of wide-bandgap perovskites may also bring energy barrier to expend $V_{bi}$ and block charge transport[30–32]. Interlayers with strong electric dipole moment have been shown to optimize the interfacial energy band structure, enhance $V_{bi}$, and improve charge transport and collection in organic and pero-SCs[33–37]. It would be highly intriguing to design a dipole interlayer that is capable of controlling the interfacial energy band structure while simultaneously passivates the defects.

In this study, we have designed a fullerene electrolyte (PCBB-3N-3I) dipole interlayer to simultaneously passivate charged surface defects and reconfigure interfacial energy band structure between perovskite active layer and electron-transporting layer (ETL). The positively charged surface defects may act as anchor sites for iodide of PCBB-3N-3I via electrostatic interaction, which can be passivated efficiently and further provide a driving force for PCBB-3N-3I assembly with preferred orientation. As a result, PCBB-3N-3I dipole interlayer reconfigures the interfacial energy band structure, resulting in enhanced $V_{bi}$ and charge collection. By treating perovskite with PCBB-3N-3I, the inverted structure planar heterojunction pero-SCs have showed simultaneously enhanced $V_{oc}$, short-circuit current density ($J_{sc}$), and fill factor (FF), leading to significantly enhanced PCE from 17.7% to 21.1%. In addition, ambient stability of PCBB-3N-3I device has also been improved.

## Results

**Device performance.** As shown in Fig. 1a, PCBB-3N-3I is the iodide quaternized derivative of a fullerene derivative terminated with tris(dimethylamine) (PCBB-3N), which exhibits high conductivity and strong molecular electric dipole moment. The inverted structure planar heterojunction pero-SCs were constructed with a structure of indium tin oxide (ITO)/poly(triarylamine) (PTAA)/MAPbI$_3$/phenyl-C$_{61}$-butyric acid methyl ester (PCBM)/Al with and without PCBB-3N-3I/PCBB-3N treatment on MAPbI$_3$ as detailed in the Methods section. Here, a small amount of excess PbI$_2$ was adopted, which gave rise to obvious PbI$_2$ peak in the X-ray diffraction (XRD) spectra of perovskite film (Supplementary Fig. 1)[38]. The excess PbI$_2$ could passivate the perovskite film and facilitate charge transportation, thus resulting in superior device performance and mitigated hysteresis[39]. The MAPbI$_3$ films showed little difference in scanning electron microscopy (SEM) images (Supplementary Fig. 2), XRD patterns (Supplementary Fig. 1), and absorption spectra (Supplementary Fig. 3) before and after coating PCBB-3N-3I/PCBB-3N, indicating that fullerene derivative treatment had little effect on morphology and crystal structure of perovskite. After coating PCBB-3N-3I or PCBB-3N layer, no additional treatment was carried out before coating the PCBM ETL.

The device was very sensitive to the concentration of PCBB-3N-3I or PCBB-3N (Supplementary Tables 1 and 2), where an extremely low concentration (0.1 mg mL$^{-1}$) gave the superior device performance. For this optimized concentration, the resulting PCBB-3N-3I or PCBB-3N film on the rough perovskite surface appears ultrathin or could even be discontinuous. Figure 1b shows the $J$–$V$ curves of champion control device (without PCBB-3N-3I/PCBB-3N treatment), PCBB-3N-3I device (with 0.1 mg mL$^{-1}$ PCBB-3N-3I treatment), and PCBB-3N device (with 0.1 mg mL$^{-1}$ PCBB-3N treatment) under AM 1.5 G illumination at a scan rate of 0.02 V s$^{-1}$ for both forward and reverse scanning in a N$_2$-filled glove box. The statistics of the photovoltaic parameters of over 20 devices for each structure are summarized in Fig. 1c and Table 1. All the devices exhibited small SDs, indicating good device reproducibility. No significant hysteresis was seen in all the devices at different scan rates (Supplementary Fig. 4). For the control devices, PCEs obtained under reverse scanning were 17.20% ± 0.22%, which lie at the baseline of state-of-the-art efficiency of inverted structure planar heterojunction pero-SCs[13,20]. Excitingly, the PCEs of PCBB-3N-3I devices significantly increased to be 20.57% ± 0.27%, owing to the simultaneously improved $V_{oc}$, $J_{sc}$, and FF. The champion PCBB-3N-3I device could even reach a PCE of 21.10% with a $V_{oc}$ of 1.105 V, a $J_{sc}$ of 23.46 mA cm$^{-2}$, and an FF of 81.36% under reverse scanning. As far as we know, this is one of the fewest inverted structure planar heterojunction pero-SCs with a PCE over 21% (Supplementary Table 3). In comparison, the PCEs of PCBB-3N devices decreased to be 15.28% ± 0.32% accompanied with overall degraded photovoltaic parameters. The device performance with a mask was also investigated, which showed little difference than that without a mask and corroborated the reliability of measurements (Supplementary Fig. 5 and Supplementary Table 4). The change in device performance with PCBB-3N-3I/PCBB-3N is also evidenced by extracted series resistance and shunt resistance as seen in Table 1. The series resistance is smaller in PCBB-3N-3I device and larger in PCBB-3N device compared with that in the control device. Figure 1d is the external quantum efficiency (EQE) spectra of the champion devices, which shows significant improvement with PCBB-3N-3I and severe deterioration with PCBB-3N. The integrated $J_{sc}$ from the EQE spectra were 21.56, 22.96, and 20.79 mA cm$^{-2}$ for the control, PCBB-3N-3I, and PCBB-3N devices, respectively, agreeing well with the $J_{sc}$ values obtained from the $J$–$V$ curves within 5% deviation. This sequence in PCEs was further confirmed by their corresponding stabilized PCEs at a maximum power point (MPP) (Supplementary Fig. 6).

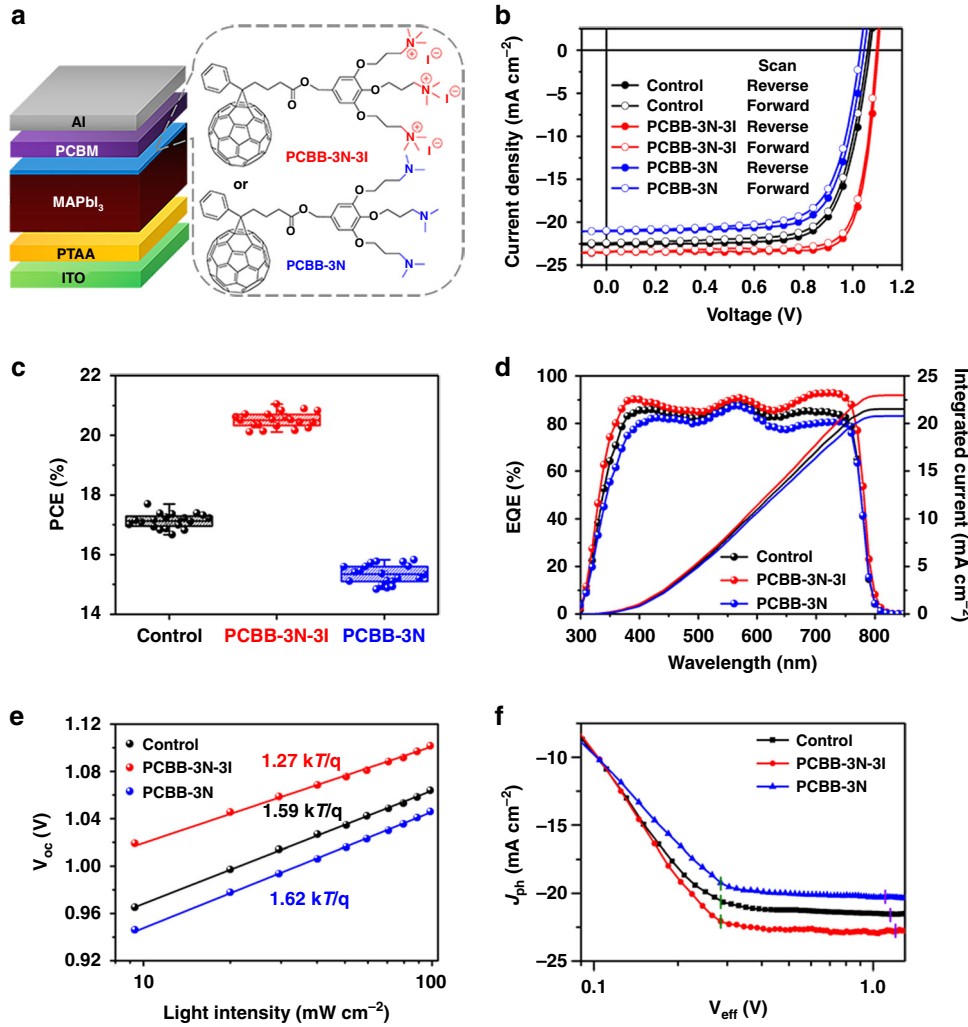

**Fig. 1** Device performance of pero-SCs. **a** Schematic illustration of layer-stacking of pero-SCs with a structure of ITO/PTAA/MAPbI$_3$/PCBM/Al with and without PCBB-3N-3I/PCBB-3N treatment on MAPbI$_3$. **b** J–V curves of champion devices under AM 1.5 G illumination at a scan rate of 0.02 V s$^{-1}$ for both forward (0 to 1.2 V) and reverse (1.2 to 0 V) scanning. **c** Statistics of PCEs for control, PCBB-3N-3I, and PCBB-3N devices. The error bars are SD in PCE of 20 devices for each structure. **d** EQE spectra and integrated current of champion devices. **e** Light intensity-dependent $V_{oc}$ for pero-SCs. **f** $J_{ph}$ vs. $V_{eff}$ curves of the pero-SCs

**Table 1 Photovoltaic parameters of pero-SCs**

|  | Scan | $V_{oc}$ (V) | $J_{sc}$ (mA cm$^{-2}$) | FF (%) | PCE (%) | $R_s$ (Ω cm$^2$) | $R_{sh}$ (Ω cm$^2$) |
|---|---|---|---|---|---|---|---|
| Control | Reverse | 1.068 | 22.61 | 73.30 | 17.70 (17.20 ± 0.22) | 1.11 | 1078 |
|  | Forward | 1.059 | 22.41 | 73.11 | 17.35 (16.89 ± 0.26) | 1.19 | 1055 |
| PCBB-3N-3I | Reverse | 1.105 | 23.46 | 81.36 | 21.10 (20.57 ± 0.27) | 0.48 | 2241 |
|  | Forward | 1.100 | 23.38 | 80.57 | 20.72 (20.12 ± 0.29) | 0.55 | 2130 |
| PCBB-3N | Reverse | 1.046 | 21.05 | 71.65 | 15.77 (15.28 ± 0.32) | 1.41 | 982 |
|  | Forward | 1.031 | 20.98 | 71.31 | 15.43 (15.00 ± 0.33) | 1.50 | 973 |

The effects of PCBB-3N-3I/PCBB-3N treatment on device performance are worth investigating. Light intensity-dependent $V_{oc}$ curves were employed to evaluate the effect of trap-assisted recombination on the devices (Fig. 1e). In the open-circuit condition, $V_{oc}$ can be described by $V_{oc} = \frac{nkT}{q}\ln(\frac{J_{ph}}{J_0} + 1)$, where $n$ is the ideality factor representing the rate of charge carrier recombination, $k$ is the Boltzmann constant, $T$ is the temperature, $J_{ph}$ is the photocurrent density, and $J_0$ is the reversed saturate current density. As $J_{ph}$ is linearly correlated to light intensity (Supplementary Fig. 7), logarithmic light intensity-dependent $V_{oc}$

can be linear fitted, which shows slopes of 1.59 k$T$/q for the control device, 1.27 k$T$/q for the PCBB-3N-3I device, and 1.62 k$T$/q for the PCBB-3N device. The smaller $n$-value of the PCBB-3N-3I device means a lower trap-assisted recombination than those of the other two devices, whereas the similar $n$-values of the control and PCBB-3N devices indicate comparable trap-assisted recombination[40,41]. The difference in $n$ of these devices was further corroborated by trap density measurement in Supplementary Fig. 8[13,17]. The trap density in PCBB-3N-3I device was lower than that in the control device and PCBB-3N device, indicating reduced surface defects of perovskite.

Figure 1f shows net photocurrent density ($J_{ph}$) vs. effective voltage ($V_{eff}$) curves under AM 1.5 G illumination of the control, PCBB-3N-3I, and PCBB-3N devices, where $J_{ph} = J_L − J_d$ ($J_L$ is the current density under AM 1.5 G illumination, $J_d$ is the dark current density) and $V_{eff} = V_0 − V$ ($V_0$ is the voltage at which $J_{ph} = 0$, $V$ is the applied bias voltage). As the total amount of photogenerated charge carriers is the same for pero-SCs before and after PCBB-3N-3I/PCBB-3N treatment, $J_{ph}$ vs. $V_{eff}$ curves illustrated the dependence of collected photogenerated charge carriers on internal electric field[33,41]. In short-circuit condition (SCC), where $V = 0$ V, the control, PCBB-3N-3I, and PCBB-3N devices showed various $V_{eff}$ of 1.15, 1.20, and 1.11 V, respectively, suggesting fullerene derivative treatment on perovskite could generate different $V_{bi}$ that are consistent with that measured by the Mott–Schottky plot (Supplementary Fig. 9). As a result, the higher $V_{eff}$ for the PCBB-3N-3I device facilitated charge transport and collection, leading to larger $J_{ph}$ and $J_{sc}$. In contrast, the PCBB-3N device with lower $V_{eff}$ gave smaller $J_{ph}$ and $J_{sc}$. It is noted that the change in $V_{bi}$ was in accordance with the $V_{oc}$ results obtained from the J–V curves, suggesting that the enhanced $V_{bi}$ could lead to a larger upper limit of $V_{oc}$[33,42]. In MPP, $V_{eff}$ of the control, PCBB-3N-3I, and PCBB-3N devices were all about 0.28 V, i.e., the internal electric field was relatively low. The ratio between $J_{ph}$ at MPP and that at SCC is 0.95, 0.97, and 0.94 for control, PCBB-3N-3I, and PCBB-3N devices, respectively, indicating different charge collection probability in MPP. As shown in Supplementary Fig. 10, the more efficient charge collection by PCBB-3N-3I treatment is consisted with lower photoluminescence (PL) intensity, leading to a higher FF[43], whereas PCBB-3N device with lower charge collection probability gives rise to higher PL intensity and a lower FF. Overall, the reduced trap-assisted recombination, enhanced $V_{bi}$, and improved charge collection of the PCBB-3N-3I device have led to significant improvement in photovoltaic parameters.

**Interfacial molecular ordering**. The drastically different effects of PCBB-3N-3I and PCBB-3N indicate that the quaternized amine with iodide counter ion plays a critical role in the interlayer, which provides a unique functionality other than fullerene moiety and tris(dimethylamine) substitution. The Lewis acid, i.e., fullerene moiety, is known to passivate negatively charged defects such as $PbI_3^−$ and under-coordinated $I^−$, but not positively charged defects such as under-coordinated $Pb^{2+}$[12,13]. Thus, the iodide in PCBB-3N-3I is proposed to bind under-coordinated $Pb^{2+}$ via electrostatic interaction, which is supported by density functional theory (DFT) calculations (Supplementary Fig. 11 and Supplementary Note 1). It reveals that PCBB-3N-3I exhibits the lowest adsorption energy (−1.20 eV) on under-coordinated $Pb^{2+}$ sites compared with other sites on the surface of perovskite. In addition, the Pb 4f peak of perovskite shifted towards higher binding energy after PCBB-3N-3I treatment in X-ray photoelectron spectroscopy (XPS) (Supplementary Fig. 12), which supported the interaction between $I^−$ and $Pb^{2+}$. These results corroborate the notion that PCBB-3N-3I can efficiently passivate positively charged surface defects and reduce trap density. In contrast, PCBB-3N showed a much higher adsorption energy (−0.62 eV) on under-coordinated $Pb^{2+}$, indicating weaker binding between the lone pair electrons of PCBB-3N and under-coordinated $Pb^{2+}$, and less efficient passivation of positively charged surface defects. Therefore, the trap density of the control and PCBB-3N devices are nearly the same.

The ionic nature of the quaterized amine iodide gives rise to a giant molecular electric dipole moment as evidenced by the DFT calculated value of 31.31 D for PCBB-3N-3I in comparison with 4.86 D for PCBM and 5.68 D for PCBB-3N (Supplementary

Fig. 13). However, whether the molecular electric dipole moment is brought together to form an interfacial electric dipole layer is critically dependent on the ordering of molecules in the ultrathin film formed on the surface of the perovskite. A completely random orientation scenario would lead to no interfacial dipole and a thin film with preferred molecular orientation may lead to a significant interfacial dipole[44].

To investigate molecular ordering in the fullerene derivative layer on the surface of perovskite, sum-frequency generation (SFG) vibrational spectrum has been employed, which is a second-order nonlinear vibration spectroscopy that is only sensitive to surface and interface structures[45]. The SFG spectra were collected from an ssp polarization combination, i.e., s-SFG, s-visible, and p-infrared (IR), and the spectral intensity was proportional to the IR dipole moment of vibration modes with a component perpendicular to the substrate[46]. As shown in Fig. 2a, $MAPbI_3$ films treated with PCBM, PCBB-3N, or PCBB-3N-3I, all showed two peaks at around 1463 $cm^{−1}$ and around at 1430 $cm^{−1}$, which correlated well with the absorption in Fourier-transform IR spectroscopy and Raman shift (Supplementary Fig. 14). The 1463 $cm^{−1}$ peak was assigned to the vibration mode of symmetry breaking in fullerene derivatives with an IR dipole moment pointing from the center of $C_{60}$ to the pendant group, which was perpendicular to the corresponding IR dipole moment of vibration mode of $C_{60}$ at around 1430 $cm^{−1}$[46]. It was found that the peak magnitude at around 1463 $cm^{−1}$ was approximately three times larger than that at around 1430 $cm^{−1}$ for PCBB-3N-3I, indicating the preferred molecular orientation on perovskite with complicated surface defects and serious morphology fluctuation. The SFG spectra at the C–H stretch region was further investigated, which corresponded to vibration modes for pendent group. The methylene symmetric stretching mode (around 2840 $cm^{−1}$), methyl symmetric stretching mode (around 2875 $cm^{−1}$) and their corresponding methylene Fermi resonance mode (around 2930 $cm^{−1}$), methyl Fermi resonance mode (around 2956 $cm^{−1}$) were visible (Fig. 2b), which further consolidate the assembly of PCBB-3N-3I with the preferred molecular orientation.

Combined with results of DFT calculation, XPS, and SFG spectra, the arrangement of PCBB-3N-3I on perovskite is illustrated in Fig. 2c. It is proposed that the iodides in PCBB-3N-3I, which are bound to under-coordinated $Pb^{2+}$ sites, may provide a driving force for molecular assembly leading to ordered packing and preferred molecular orientation. The assembly of PCBB-3N-3I forms a dipole interlayer with its negative end pointed towards perovskite and positive end pointed outside (black arrow), which is aligned with the direction of built-in field of the pero-SC devices. The superposition of the built-in field and the interfacial dipole results in a greater local electric field, which, in combination with reduced interfacial potential energy loss due to lower trap density, constitute the major factors that contribute to the benefits of PCBB-3N-3I treatment, i.e., enhanced $V_{bi}$ and improved charge collection efficiency.

In contrast, perovskite treated with PCBB-3N or PCBM showed two peaks at around 1430 $cm^{−1}$ and around 1463 $cm^{−1}$ with comparable intensity, indicating random orientation of the molecules (Fig. 2d). The deteriorated $V_{bi}$ and charge collection with PCBB-3N treatment can be explained by impediment of charge transport, which is attributed to the insulating long side chain of PCBB-3N, as evidenced by lower conductivity and mobility (Supplementary Fig. 15 and Supplementary Table 5).

**Interfacial energy band structure**. The SFG spectra measurements establish the presence of strong molecular ordering in the PCBB-3N-3I interlayer, but the hypothesized mechanism of improved $V_{bi}$ and charge collection needs more experimental

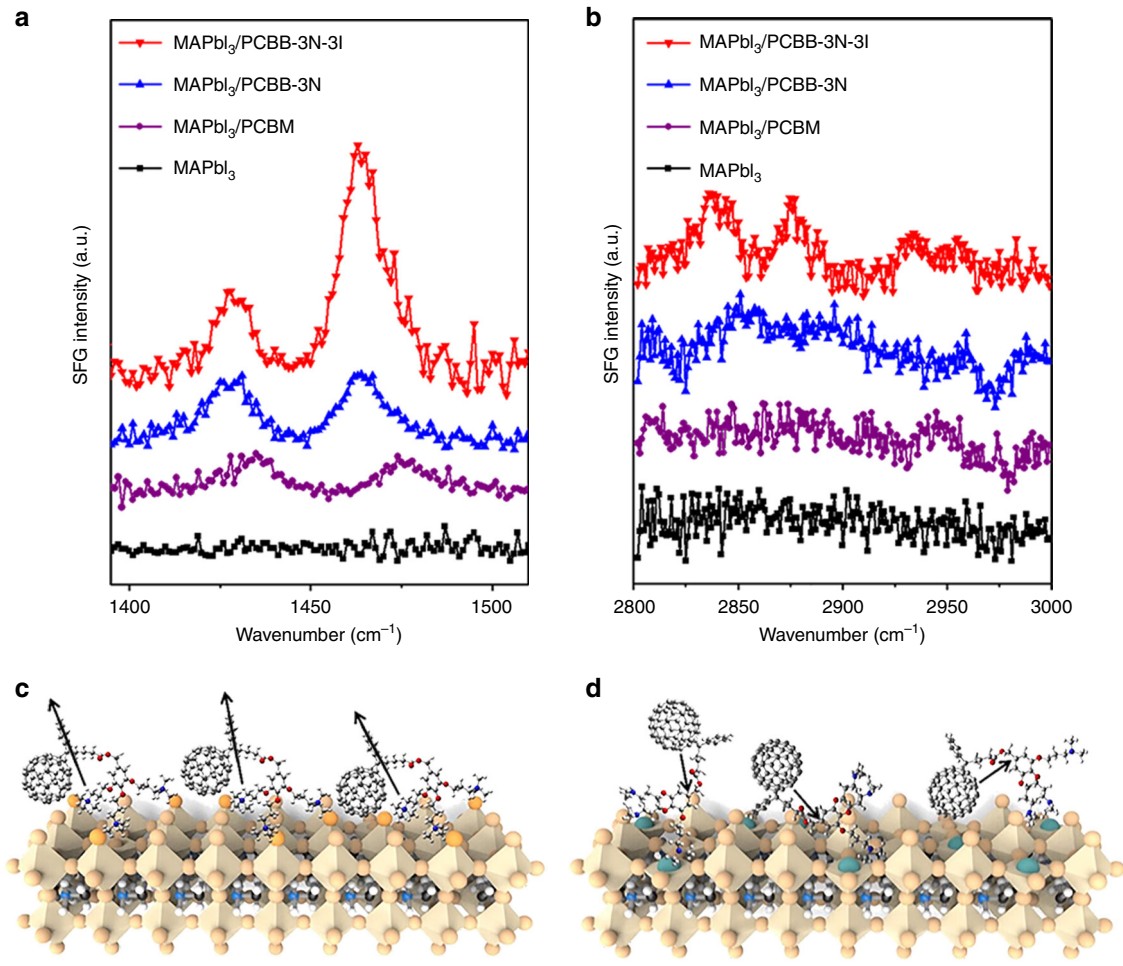

**Fig. 2** The arrangement of fullerene derivatives on the perovskite film. **a**, **b** SFG spectrum of perovskite before and after treatment with PCBM, PCBB-3N-3I, and PCBB-3N at **a** region corresponding to vibration modes for $C_{60}$ moiety and **b** C–H stretch region corresponding to vibration modes for pendant group. **c**, **d** Schematic illustration of molecular orientation of **c** PCBB-3N-3I and **d** PCBB-3N on $MAPbI_3$

evidence. Operando scanning Kelvin probe microscopy (SKPM) on cross-sections of these pero-SCs has been employed to investigate interfacial energy band alignment and analyze the mechanism of improved performance[47–52]. To obtain high-quality cross-sections with smooth morphology and clear layer contrast, the exposed edge of crosscut devices was ion-milled under an $Ar^+$ beam following the procedure published in our previous work[49]. Figure 3a shows the cross-sectional SEM image of the control device; all functional layers can be clearly distinguished. Importantly, this cross-section could also be successfully characterized with atomic force microscopy (AFM), which showed small surface roughness of a few tens of nanometers in the topography image (Fig. 3b) and clear mechanical property contrast between the layers in the phase image (Fig. 3c). The corresponding height and phase profiles were shown in Supplementary Fig. 16. As seen from the $J–V$ curves in Supplementary Fig. 17 and the photovoltaic parameters in Supplementary Table 6, the crosscut devices can effectively maintain the photovoltaic properties after SKPM measurement. The robust crosscut device warrants the reliability of the operando SKPM measurements.

Cross-sectional SKPM measurements were carried out to detect surface potential (SP) images of the control device in SCC (Fig. 3d). The corresponding SP profile averaged over five slow-scan lines was extracted for Fig. 3e. The SP changed slightly across the perovskite film but dramatically dropped at both the

PTAA/perovskite and perovskite/PCBM interfaces, which was attributed to $n$-type doping in perovskite based on ultraviolet photoelectron spectroscopy (UPS) (Supplementary Fig. 18) that narrows depletion layer width. Moreover, it is interesting to see that the perovskite treated with PCBB-3N-3I/PCBB-3N brought significant changes in the potential drop at the perovskite/PCBM interface. For example, compared with the control device, PCBB-3N-3I (Supplementary Fig. 19 and Fig. 3e) increased the potential drop; in contrast, PCCB-3N reduced it (Supplementary Fig. 20 and Fig. 3e). We also noted that PCBB-3N-3I/PCBB-3N treatment had little influence on the potential across the perovskite film and at the PTAA/perovskite interface. By taking the derivative of SP profiles (Fig. 3f), the local electric field at the perovskite/PCBM interface was obtained, and that in PCBB-3N-3I-treated device is clearly stronger than the control device, which is stronger than that in PCBB-3N-treated device. This result undoubtedly agrees with the postulation inferred from the ordering of molecular structure at the interface.

In SCC with an equilibrium state, the Fermi level was aligned across the device and the vacuum level ($E_{vac}$) offsets were established to compensate for the work function difference between adjacent layers. To understand the effect of PCBB-3N-3I or PCBB-3N on the cathode interfacial energy band structure of the operating device, the $E_{vac}$ offsets in SCC were illustrated by multiplying the SP profiles by the elementary electron charge (Fig. 4a)[53]. For the control device, it was found that the $E_{vac}$

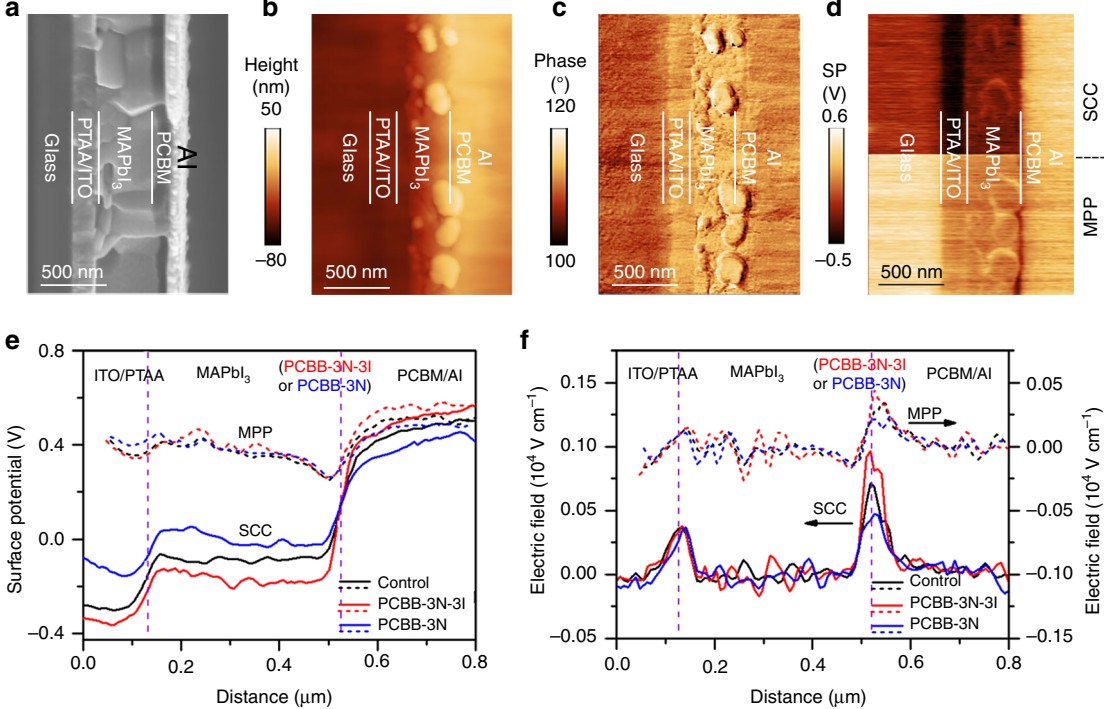

**Fig. 3** Visualization of interfacial energy depth profiles in SCC and MPP. **a–d** Control device cross-section: **a** SEM image; **b** AFM topography image; **c** AFM phase image; **d** SKPM-measured SP image in SCC (top half) and MPP (bottom half). **e** SP depth profiles and **f** corresponding electric field distribution of the control device (black), PCBB-3N-3I device (red), and PCBB-3N device (blue) in SCC (solid line) and MPP (dashed line)

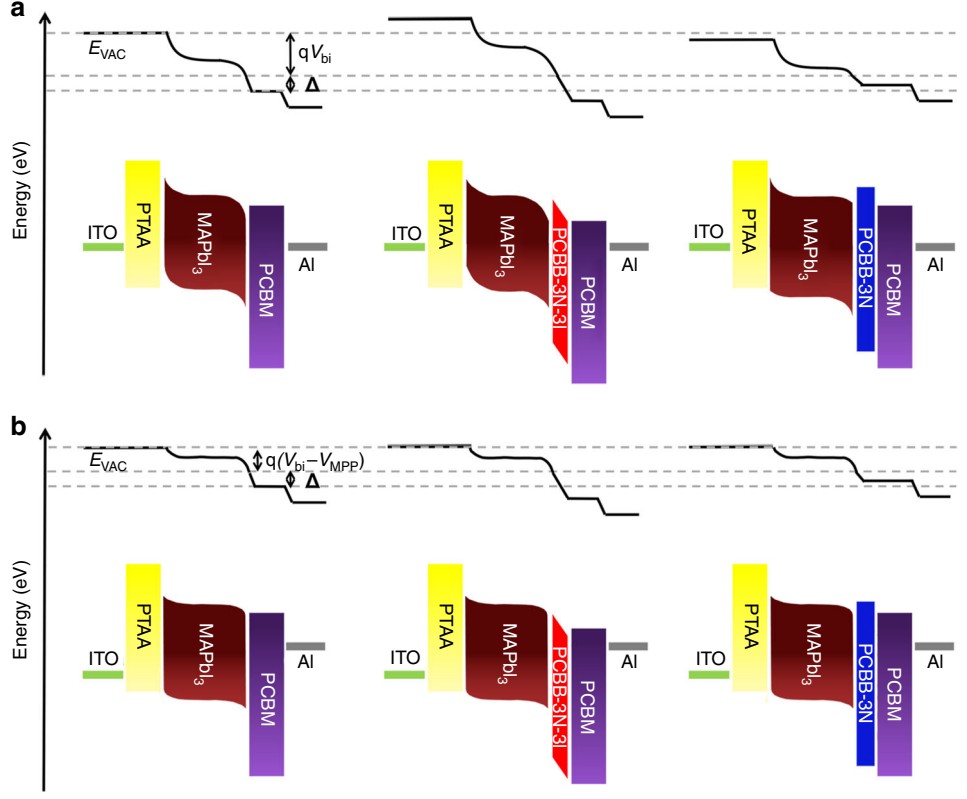

**Fig. 4** Energy band structure of pero-SCs **a** in SCC and **b** MPP. Here, energy band alignment refers to the interfacial energy level offset. Both built-in potential and interfacial dipole moment improve with PCBB-3N-3I dipole interlayer and deteriorate with PCBB-3N interlayer

offsets at the perovskite/PCBM interface consisted of both an interfacial dipole ($\Delta$) and the extended band bending of perovskite. The charge carriers injected from PCBM into the surface defect states of perovskite will generate an interfacial dipole due to charge accumulation, which inevitably generated interfacial potential energy loss to weaken the band bending of the perovskite[21,22]. Interestingly, after treating MAPbI$_3$ with PCBB-3N-3I, the band bending of perovskite near PCBM was enlarged. It demonstrated that more efficient passivation of perovskite surface defects by PCBB-3N-3I will lower accumulated charges at the perovskite/PCBM interface, leading to reduced interfacial potential energy loss. In contrast, the band bending of perovskite near PCBM was weakened after PCBB-3N treatment. Considering the comparative passivation ability between PCBM and PCBB-3N, the weakened band bending of perovskite could be explained by blocked charge injection from PCBM towards the perovskite interior due to the insulating long side chain of PCBB-3N. In inverted structure planar heterojunction devices, the $V_{bi}$ can be defined as the potential difference across the perovskite active layer[54]. The enlarged band bending of perovskite in the PCBB-3N-3I device suggested a higher $V_{bi}$ than that of the control device, which is consistent with the Mott–Schottky measured values, thus contributing to higher $V_{oc}$, $J_{sc}$, and EQE.

Furthermore, cross-sectional SKPM was carried out to detect SP images of all three devices in MPP (Fig. 3d). During this measurement, forward bias voltage ($V_{MPP}$) equal to the voltage at MPP was applied between electrodes, giving an electric field with a reversed direction to that of built-in field, which lowered the internal electric field of perovskite as supported by the derivatives of SP profiles in Fig. 3f. Interestingly, all three devices exhibited similar band bending of perovskite in Fig. 3e, which was consisted with similar $V_{eff}$ at MPP in Fig. 1f. However, the SP drop and related electric field at the perovskite/PCBM interface enlarged in PCBB-3N-3I device but reduced in PCBB-3N device compared with that of control device, indicating significant difference in interfacial dipole as schematic in energy band structure in MPP (Fig. 4b). Based on UPS, the $E_{vac}$ downshifts were 0.06, 0.19, and 0.02 eV for perovskite with PCBM, PCBB-3N-3I, and PCBB-3N treatment, respectively (Supplementary Fig. 18), which agreed with the interfacial dipole obtained by cross-sectional SKPM. The larger downshift in $E_{vac}$ with the assembled PCBB-3N-3I suggests the formation of a dipole interlayer with the negative end pointed towards perovskite and the positive end pointed outside as proposed in Fig. 2c, which brings significant change in interfacial energy band structure, leading to a downshift in lowest unoccupied molecular orbital (LUMO) of PCBB-3N-3I to lower charge collection barrier.

As the direction of electric dipole moment of PCBB-3N-3I dipole interlayer aligns with that of interfacial dipole induced by charge injection from PCBM to the surface defect states of perovskite, the interfacial dipole will be reinforced as a result of the superposition[33]. Even if $V_{eff}$ had little difference in MPP for all three devices, the enlarged interfacial dipole in PCBB-3N-3I device improved charge collection, leading to significantly enhanced FF.

**Device stability**. The normalized PCE evolution vs. storage time was evaluated by employing the device structure of ITO/PTAA/MAPbI$_3$/PCBM/C$_{60}$/bathocuproine(BCP)/Cu with or without PCBB-3N-3I/PCBB-3N. It is noted that the Cu and C$_{60}$/BCP bilayer ETL were used instead of Al to restrain reactivity between the cathode/perovskite film[55]. Then, as shown in Fig. 5a, the devices (over five devices for each structure) stored in ambient condition with 40–50% relative humidity (RH) experienced a rapid exponential decay at the first 150 h and maintained 85%,

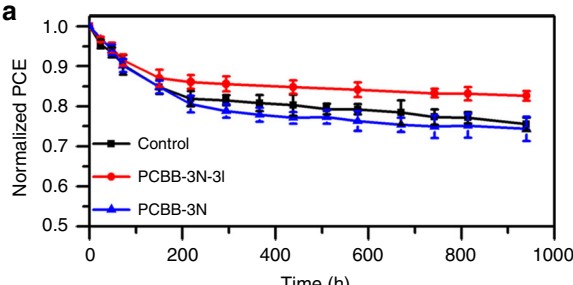

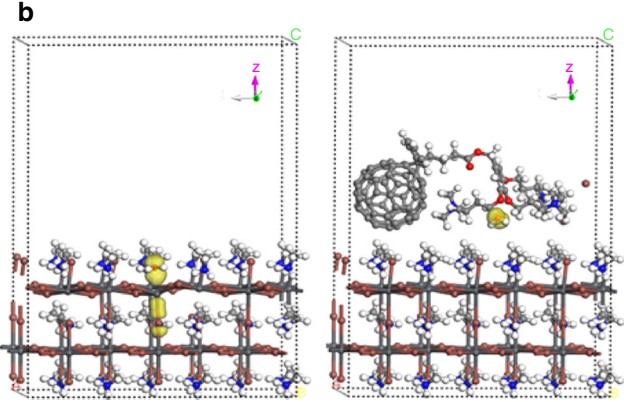

**Fig. 5** Device stability of pero-SCs. **a** Normalized PCE of devices with a structure of ITO/PTAA/MAPbI$_3$/PCBM/C$_{60}$/BCP/Cu with and without PCBB-3N-3I/PCBB-3N treatment on MAPbI$_3$ as a function of storage time in ambient condition with 40–50% RH. The error bars are SD in PCE of five devices for each structure. **b** DFT calculation of interaction between $p$-orbital of O$^{2-}$ of adsorbed H$_2$O and $p$-orbital of Pb$^{2+}$ of MAPbI$_3$ before and after PCBB-3N-3I treatment

87%, and 85% of initial PCEs in the control devices, PCBB-3N-3I devices, and PCBB-3N devices, respectively. The rapid exponential decay is also called as "burn-in," which may arise from interfacial degradation due to adsorption of H$_2$O and/or O$_2$ on fullerene interlayer[56]. Nevertheless, the PCBB-3N-3I devices still maintained 83% of its initial PCE after 940 h, which were better than the control devices and PCBB-3N devices. Even in ambient condition with 75–85% RH, the PCBB-3N-3I devices maintained 62% of its initial PCE after 500 h, which also surpassed control devices and PCBB-3N devices in stability (Supplementary Fig. 21). The improved ambient stability of PCBB-3N-3I devices can be explained by reduced surface defects[13,19]. Based on DFT calculation, adsorption energy of H$_2$O and O$_2$ increased after treating by PCBB-3N-3I, which is attributed to weakened interaction between the $p$-orbital of O$^{2-}$ in adsorbed H$_2$O and the $p$-orbital of under-coordinated Pb$^{2+}$ in perovskite (Fig. 5b). In addition, the excellent ambient stability of PCBB-3N-3I itself also played a positive role (Supplementary Fig. 22).

## Discussion

Our work opens up a possibility for material and interface design based on the charged surface defects in pero-SCs. In some cases, the detrimental charged defect states at the surface of the perovskite active layer can provide an entry point to induce assembly of the sequent interlayer, leading to reconfigure interfacial energy band structure to enhance $V_{bi}$ and charge collection. This work also reveals the critical role of cross-scale characterizations in device interface research. The microscopic chemical structure and molecular orientation at the interface are directly correlated with macroscopic device performance via nano- and mesoscopic interface dipole and energy band structure measurements. We

believe similar in-operando and in-situ cross-scale characterizations will be tremendously beneficial to understanding and exploring interfacial effects in future thin-film electronic and optoelectronic devices.

In conclusion, treating perovskite active layer by Lewis-acid-featured fullerene skeleton after iodide ionization (PCBB-3N-3I) can reduce trap-assisted recombination and improve $V_{bi}$ and charge collection simultaneously, leading to superior device performance of inverted structure planar heterojunction pero-SCs. It is proposed that positively charged defects on the surface of perovskite will provide binding sites for iodide, which can be passivated efficiently and further promote assembly of PCBB-3N-3I to form a dipole interlayer, resulting in optimized interfacial energy band structure. In contrast, the device performance deteriorates by treating with PCBB-3N before iodide ionization, which exhibits random molecular orientation and blocks charge transport, leading to deteriorated interfacial energy band structure. It is demonstrated that rational material and interface design according to the specific surface chemical structure is highly promising in tuning molecular arrangement and energy band structure, resulting in superior device performance and stability.

## Methods

**Materials**. $PbI_2$ (99.999%) was purchased from Alfa Aesar. MAI (99%) was purchased from Shanghai Mater. PTAA was purchased from Xi'an Polymer Light Technology Corp. The 2,3,5,6-Tetrafluoro-7,7,8,8-tetracyanoquinodimethane (F4-TCNQ) was purchased from Jilin OLED company. PCBM was purchased from Solarmer Materials, Inc. All the materials were used as received. PCBB-3N-3I and PCBB-3N were prepared based on our previous work[57].

**Device preparation**. ITO glass substrates (15 Ω sq$^{-1}$) were sequentially washed with isopropanol, acetone, distilled water, and ethanol by ultrasonication for 20 min. PTAA containing 1 wt% F4-TCNQ (2 mg mL$^{-1}$ in toluene) was spin-coated at 5000 r.p.m. for 30 s onto the ITO glass substrate, followed by annealing at 100 °C for 10 min. Then, 35 μL perovskite precursor solution ($PbI_2$:MAI with a molar ratio of 1.3:0.3 dissolved in dimethyl formamide:dimethyl sulfoxide (DMF:DMSO) with a volume ratio of 9:1) was spin-coated at 6000 r.p.m. for 15 s, followed by spin-coating 40 μL MAI (35 mg mL$^{-1}$ in IPA) at 4000 r.p.m. for 45 s and annealing at 100 °C for 30 min. The PCBB-3N-3I or PCBB-3N (0–0.5 mg mL$^{-1}$ in tri-fluoroethanol) was spin-coated onto the perovskite at 6000 r.p.m. for 30 s. After that, PCBM (20 mg mL$^{-1}$ in chlorobenzene) was spin-coated at 2000 r.p.m. for 30 s. Finally, Al (100 nm) or $C_{60}$ (20 nm)/BCP (8 nm)/Cu (100 nm) were thermal-evaporated onto PCBM under $1 \times 10^{-6}$ mbar. The active area of device is 0.1 cm$^2$, which is defined by cross-overlap of the 0.2 cm wide-patterned ITO bar and the 0.5 cm wide Al/Cu bar deposited through a shadow mask. To corroborate the reliability of device measurements, non-reflective mask with an area of 0.049 cm$^2$ was also used to define the cell area. The device cross-section was fabricated using an Ilion$^+$ 697 System (Gatan, Inc.) based on our previous report[49]. The device was mechanically cleaved in $N_2$-filled glove box and mounted into the milling chamber with a vacuum transfer box to avoid contamination. After cooling by liquid nitrogen, the exposed edge of the device was milled by Ar$^+$ beam (beam voltage 5 keV, beam current 10 μA) under about $6 \times 10^{-5}$ Torr for 2 h.

**Characterization**. The SEM images were collected on Hitachi SU8010. The XPS/UPS was measured by PHI ULVAC-5000 VP III. The PL was tested by Edinburgh Instrument FLS980. The SFG system was built by EKSPLA: the visible beam (incident angle 60°, 532 nm) and IR beam (incident angle 55°, around 1360–1550 cm$^{-1}$) were about 25 ps at 50 Hz with energy <200 mJ. Mott–Schottky plot were measured using Zahner Ennium Electrochemical Workstation. The $J$–$V$ characteristics of pero-SCs were measured using a Keithley 2400 source meter under AM 1.5 G illumination (100 mW cm$^{-2}$) from a SS-F5-3A solar simulator (Enli Technology, Co., Ltd) without any preconditioning. The light intensity was calibrated by a standard Si solar cell (SRC-00036, Enli Technology Co., Ltd). The spectral mismatch factor between the standard Si solar cell and our devices was 1.017. The EQE spectra were obtained using a QE-R3011 solar cell spectral response measurement system (Enli Technology, Co., Ltd). The light intensity at each wavelength was also calibrated with a standard Si solar cell (SRC-00036). The frequency-modulation (FM) SKPM was operated combined with a Cypher S AFM (Asylum Research, Oxford Instruments) and a HF2LI Lock-in amplifier (Zurich Instruments) in $N_2$-filled glove box. The resonance frequency $\omega_0$ and spring constant of AFM conducting tips are around 140 kHz and 5.0 N m$^{-1}$, respectively. For FM-SKPM measurement, standard AC mode imaging was performed to acquire the topography; meanwhile, an AC voltage (typically 2 V in amplitude and 1 kHz in frequency $\omega_e$) was applied and the DC voltage applied by Kelvin controller to nullify sideband at $\omega_0 \pm \omega_e$ was collected as SP. For SCC, the Al electrode and the ITO electrode was directly connected; for MPP, forward bias $V_{MPP}$ was applied via a tunable voltage source between the electrodes.

**Reporting summary**. Further information on research design is available in the Nature Research Reporting Summary linked to this article.

## Data availability

The datasets generated during and/or analysed during the current study are available from the corresponding author on reasonable request.

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

## Acknowledgements

This work was supported by the Ministry of Science and Technology of China (Grant Number 2016YFA0200700), National Natural Science Foundation of China (Grant Numbers 51922074, 51673138, 51820105003, 91633301, 21625304, and 21875280), Priority Academic Program Development of Jiangsu Higher Education Institutions (PAPD), the Jiangsu Provincial Natural Science Foundation (Grant Numbers BK20160059), Collaborative Innovation Center of Suzhou Nano Science and Technology, and Collaborative Innovation Center for New-type Urbanization and Social Governance of Jiangsu Province. We are grateful for the technical support for UPS measurements from Nano-X, SINANO. We acknowledged Dr. Jiandong Zhang for DFT calculation and Dr. Romain Stomp, Wei Yu for helpful discussion on SKPM setup.

## Author contributions

Y.F.L., L.C., Y.W.L. and Q.C. conceived the idea, designed the experiments, and wrote the manuscript. Q.C., C.W. and J.L. performed cross-sectional SKPM and XPS/UPS. J.Y. and H.L. measured SFG. J.L.Y. measured Raman spectra. R.X. synthesized PCBB-3N-3I and PCBB-3N. M.Z. and Y.Z prepared pero-SCs and performed all other measurements. All of the authors commented on the manuscript.

## Competing interests

The authors declare no competing interests.
