## [Peer Review File · Nature Communications]

Reviewers' comments:

Reviewer #1 (Remarks to the Author):

In this work, the authors report new defect passivation method using fullerene derivative. As the layer is inserted between the PCBM and the perovskite layer, it provides better energy alignment in the device and, more importantly, simultaneously passivates the PCBM layer and the perovskite layer.

Overall, the authors performed a series of experiments to prove their strategy, but some points should be included.

Therefore, I recommend its publication in Nature Communications, with major modifications.

1. The exact behavior of materials between the PCBM layer and the perovskite layer is highly important for effective passivation. Therefore, in addition to the DFT calculation results, additional evidence from direct observation between PCBM/PCBB-3N-3I and PCBB-3N-3I/MAPbI₃ should be considered. For example, IR shift and Raman shift between two layers can be monitored for clarity.

2. Material stability of PCBB-3N-3I itself should be tested and proved, because it is an ionic compound. The authors have performed device stability test, but the relative humidity was 40-50%. If the humidity goes up to >60~85%, the stability of PCBB-3N-3I might be changed. Therefore, to be generally applied into other kinds of devices, the stability test is essential.

Reviewer #2 (Remarks to the Author):

In this paper, the authors present a new strategy for optimizing the charge collection of perovskite surfaces. They introduce a fullerene-based molecule PCBB-3N-3I which can act as dipolar layer to enhance the interfacial electric field and charge extraction efficiency of the perovskite/ETL interface. The PCBB-3N-3I molecules are self-assembled and oriented in a coherent manner, which builds a thin layer with net polarization. For comparison, another fullerene derivative (PCBB-3N) with a weak dipole moment was also used to assess the performance. Experimental evidence (Voc curves, SFG, SKPM, DFT calculation) is shown, to prove the passivation of recombination traps, the coherent orientation of PCBB-3N-3I and a resulting beneficial increase in built-in potential.

However, the photocurrent and the external quantum efficiency (EQE) of the device were obviously overestimated in this work, which significantly reduces the reliability of I-V measurement. On the other hand, the introduction of dipolar layer to enhance charge collection efficiency has been investigated in the previous work (Energy Environ. Sci. 11, 1880-1889 (2018)), which is not a very new concept. Therefore, from both novelty and device performance perspectives, I think the paper does not reach the high standard set by Nature Communications. It may more suitable for other sister journals if the following concerns are addressed.

1. The authors claimed that perovskite/PCBB-3N-3I interface showed much stronger charge collection ability than that of the perovskite/PCBB-3N and perovskite/PCBM interface due to the large dipole moment. I wonder the energy levels of PCBB-3N-3I or PCBB-3N itself also play a critical role on the interfacial charge collection efficiency in the perovskite solar cells. Thus, the authors should give more information about the HOMO and LUMO energy of the PCBB-3N-3I and PCBB-3N molecules and carefully verify the reliability of the experimental results.

2. Line 125: a typo of “standard derivations” for “standard deviations”

3. Fig 1d. shows that the EQE is increased differently according to the wavelength considered. The higher increase at high wavelength might indicate better charge extraction and reduced recombination at the back interface with PCBM which is coherent with the phenomena reported throughout the article. However, it does not appear obvious to me why the low wavelength show a higher increase in EQE, because unchanged electric field in fig. 3f suggests that the front charge extraction at the interface with PTAA is not expected to be significantly impacted. Therefore, this higher EQE increase could indicate that there is after all an effect induced on the perovskite/PTAA interface or maybe that the optical absorption properties have been modified, which might be worth discussing.

Reviewer #3 (Remarks to the Author):

“Reconfiguration of interfacial energy band structure for high-performance p-i-n perovskite solar cells”, by M. Zhang, Q. Chen, R. Xue, Y. Zhan, C. Wang, J. Lai, J. Yang, H. Lin, Y. Li, L. Chen, and Y. Li.

The authors showed that the charged surface defects can be benign after passivation and further exploited to provide favorable interfacial energy band alignment leading to enhanced PEC in perovskite solar cell. Overall the discussion is good. The SKPM and UPS techniques used to show an “experimental” band diagram can be valuable to the Solid State community as a whole. Before the paper can be accepted, the authors should address the following questions:

- Table 1: Did the authors measure or determine the device series and shunt resistances? If adding this to the table, it could provide more insight about device performance. For instance, I noticed a slightly shallower I-V curve for the device with the PCBB-3N capping layer compared to the control group (no capping layer), suggesting that the series resistance is higher when using PCBB-3N. However, the series resistance appears lower for the PCBB-3N-3I capping layer compared to the control group. This was explained later by the SKPM measurements, but the series resistance argument could corroborate this.
- Line 122: “...at a scan rate of 0.02 V s⁻¹...” Is there a reason for choosing this scan rate? Does scan rate have any impact on the device performance, as previous literature mentioned?
- Line 235: “The assembly of PCBB-3N-3I forms ... and positive end pointed outside (black arrow)...” Is there any evidence that the positive end points outward, rather than towards a negative defect (i.e. PbI₃-)? I’m not fully convinced on their claims of specific orientation of the fullerene component based on the data provided.
- Fig. 3d: The surface potential difference in the “bulk” region of the perovskite is nearly uniform, even in the short-circuit condition (SCC, top of the image). This suggests that most band bending occurs close to the interface. The schematic of Fig. 4 (derived from UPS) also shows this, which is good. This is surprising, for what should be an “intrinsic” material.
- Line 275: “The SP changed slightly across the perovskite film but dramatically dropped due to the existence of two junctions, which agrees with the behavior of p-i-n structure devices”. But in a typical p-i-n a large drift field is maintained. This comment is related to the one above.
- Line 370: “...can reduce trap-assisted recombination...”. What are the direct evidence of the reduction of trap states?
- Fig. S2: The significant presence of PbI₂ in the XRD plots is concerning. It is quite surprising that their device performance was as high as reported based on the PbI₂ impurity peak. The authors should provide some comment on this.
- Fig. S5: The author’s method of Mott-Gurney analysis to measure the trap density in Perovskite

using their structures is questionable. It should be a metal-semiconductor-metal structure, unless they can argue that the effects of transport layers can be negated. If so, they should remove their trap density argument as it is not well proven by their data.

Reviewers' comments:

Reviewer #1 (Remarks to the Author):

In this work, the authors report new defect passivation method using fullerene derivative. As the layer is inserted between the PCBM and the perovskite layer, it provides better energy alignment in the device and, more importantly, simultaneously passivates the PCBM layer and the perovskite layer. Overall, the authors performed a series of experiments to prove their strategy, but some points should be included. Therefore, I **recommend its publication in Nature Communications**, with major modifications.

1. The exact behavior of materials between the PCBM layer and the perovskite layer is highly important for effective passivation. Therefore, in addition to the DFT calculation results, additional evidence from direct observation between PCBM/PCBB-3N-3I and PCBB-3N-3I/MAPbI₃ should be considered. For example, IR shift and Raman shift between two layers can be monitored for clarity.

Response: We thank reviewer for critical comment on the effect of PCBB-3N-3I. According to reviewer's suggestions, we performed Fourier transform infrared spectroscopy (FT-IR) and Raman spectra of PCBM and MAPbI₃ films with and w/o PCBB-3N-3I or PCBB-3N treatment (**Figure A**). Unfortunately, both the PCBM and MAPbI₃ shows negligible FT-IR and Raman shifts. We guess the resolution of FT-IR and Raman spectra is too low to detect possible peaks shift of perovskite and PCBM induced by the ultrathin (~ 1 nm) PCBB-3N-3I/PCBB-3N.

Figure A. a-b FT-IR and **c-d** Raman spectra of MAPbI₃ and PCBM before and after PCBB-3N-3I/PCBB-3N treatment.

In order to explore the effect of ultrathin PCBB-3N-3I treatment on MAPbI₃, X-ray photoelectron spectroscopy (XPS) was employed, which enables detects chemical structure in the near surface region. **Figure B** shows XPS spectra of MAPbI₃ before and after PCBM, PCBB-3N-3I and PCBB-3N treatment. Only Pb 4f of MAPbI₃ with PCBB-3N-3I treatment shows ~ 0.21 eV shift toward high binding energy. Combining with the DFT calculated lowest adsorption energy of iodide of PCBB-3N-3I on under-coordinated Pb²⁺ of MAPbI₃, the shift in binding energy of Pb 4f suggests a strong interaction occurred between iodide of PCBB-3N-3I and Pb of MAPbI₃, indicating efficient passivation of under-coordinated Pb²⁺ of MAPbI₃. Then, this interaction further provides a driving force for assembly of PCBB-3N-3I with preferred molecular orientation, which was confirmed by the higher peak magnitude at ~ 1463 cm⁻¹ than that at 1430 cm⁻¹ in SFG spectra at region corresponding to vibration modes for C₆₀ moiety (**Figure Ca** and **Figure 2a**). To further prove the assembly of PCBB-3N-3I, SFG spectra at C-H stretch region corresponding to vibration modes for pendant group was performed (**Figure Cb**). The methylene symmetric stretching mode (~ 2840 cm⁻¹), methyl symmetric stretching mode (~ 2875 cm⁻¹) and their corresponding methylene Fermi resonance mode (~ 2930 cm⁻¹), methyl Fermi resonance mode (~ 2956 cm⁻¹) were observed (Figure C(b)), which consolidate the assembly of PCBB-3N-3I with preferred molecular orientation. On the contrary, PCBB-3N-3I on Si does not show any preferred molecular orientation (**Figure Cc-d**).

Figure B. XPS for **a** C 1s, **b** N 1s, **c** Pb 4f, **d** I 3d of MAPbI₃ before and after PCBM, PCBB-3N-3I and PCBB-3N treatment.

To understand the interaction between PCBM and underlying PCBB-3N-3I, SFG spectra of PCBB-3N-3I or PCBB-3N coated by PCBM (~ 1 nm) were measured. As shown in **Figure Cc-d**, only the peak of MAPbI₃/PCBB-3N-3I/PCBM at ~ 1463 cm⁻¹ is still higher than that at ~ 1430 cm⁻¹ (corresponding to C60 moiety), and vibration modes at C-H stretch region (corresponding to pendant group) are also visible. These results demonstrate that the preferred molecular orientation of PCBB-3N-3I can be well maintained even after coating PCBM.

Figure C. SFG spectra of MAPbI₃ before and after fullerene derivative treatment. **a-b** MAPbI₃ with and without PCBM/PCBB-3N-3I/PCBB-3N at **a** region corresponding to vibration modes for C₆₀ moiety and **b** C-H stretch region corresponding to vibration modes for pendant group. **c-d** PCBB-3N-3I or PCBB-3N coated with PCBM at **c** region corresponding to vibration modes for C₆₀ moiety and **d** C-H stretch region corresponding to vibration modes for pendant group.

In response to the reviewer's critical comments, we have added the discussion in the revised manuscript in page 11-13, and the Figure A, B, Cb as Supplementary Fig. 13, 11, 2b and Fig. 2b in revised manuscript.

Page 11: "In addition, the Pb 4f peak of perovskite shifted towards higher binding energy after PCBB-3N-3I treatment in X-ray photoelectron spectroscopy (XPS) (Supplementary Fig. 11), which supported the interaction between I⁻ and Pb²⁺."

Page 12: "and Raman shift (Supplementary Fig. 13)."

Page 13: "The SFG spectra at C-H stretch region was further investigated, which corresponded to vibration modes for pendent group. The methylene symmetric stretching mode (~ 2840 cm⁻¹), methyl symmetric stretching mode (~ 2875 cm⁻¹) and their corresponding methylene Fermi resonance mode (~ 2930 cm⁻¹), methyl Fermi resonance mode (~ 2956 cm⁻¹) were visible (Fig. 2b), which further consolidate the assembly of PCBB-3N-3I with preferred molecular orientation.

Page 13: "Combined with results of DFT calculation, XPS and SFG spectra,"

2. Material stability of PCBB-3N-3I itself should be tested and proved, because it is an ionic compound. The authors have performed device stability test, but the relative humidity was 40-50%. If the humidity goes up to >60~85%, the stability of PCBB-3N-3I might be changed. Therefore, to be generally applied into other kinds of devices, the stability test is essential.

Response: We appreciate reviewer's rich experience in material stability. According to reviewer's suggestion, we kept PCBB-3N-3I in ambient condition with 75-85% RH to evaluate its intrinsic stability. As shown in **Figure D**, all the characterizations including FT-IR, time-of-flight mass spectroscopy (TOF-MS) and ¹H-NMR show little change after ~ 500 h, indicating that PCBB-3N-3I is highly stable.

We also tested the device stability with the same structure of ITO/PTAA/MAPbI₃/PCBM/C₆₀/BCP/Cu with and w/o PCBB-3N-3I/PCBB-3N in ambient condition with 75-85% RH. As shown in **Figure E**, it is found that control devices, PCBB-3N-3I devices and PCBB-3N devices maintained ~ 44%, ~ 61% and ~ 46% of their initial PCE after 500 h, respectively. Although the accelerated degradation cannot be avoided in higher RH, the lower defect density and excellent

material stability of PCBB-3N-3I itself guaranteed better stability of PCBB-3N-3I devices than that of control and PCBB-3N devices.

Figure D. Structure characterization of PCBB-3N-3I stored in ambient condition with 75-85% RH: **a** ¹H-NMR; **b** TOF-MS; **c** FT-IR.

We have added Figure D and E as Supplementary Figure 21 and 20, respectively. We have also added the discussion in page 19 as follows: “Even in ambient condition

with 75-85% RH, the PCBB-3N-3I devices maintained ~ 62% of its initial PCE after ~ 500 h, which also surpassed the control and PCBB-3N devices in stability (Supplementary Fig. 20).” and “In addition, the excellent ambient stability of PCBB-3N-3I itself also played a positive role (Supplementary Fig. 21).”

Figure E. Device stability with a structure of ITO/PTAA/MAPbI₃/PCBM/C₆₀/BCP/Cu with and w/o PCBB-3N-3I/PCBB-3N treatment in ambient condition with 75-85% RH.

Reviewer #2 (Remarks to the Author):

In this paper, the authors present a new strategy for optimizing the charge collection of perovskite surfaces. They introduce a fullerene-based molecule PCBB-3N-3I which can act as dipolar layer to enhance the interfacial electric field and charge extraction efficiency of the perovskite/ETL interface. The PCBB-3N-3I molecules are self-assembled and oriented in a coherent manner, which builds a thin layer with net polarization. For comparison, another fullerene derivative (PCBB-3N) with a weak dipole moment was also used to assess the performance. Experimental evidence (V_{oc} curves, SFG, SKPM, DFT calculation) is shown, to prove the passivation of recombination traps, the coherent orientation of PCBB-3N-3I and a resulting beneficial increase in built-in potential.

However, the photocurrent and the external quantum efficiency (EQE) of the device were obviously overestimated in this work, which significantly reduces the reliability of I-V measurement. On the other hand, the introduction of dipolar layer to enhance charge collection efficiency has been investigated in the previous work (*Energy Environ. Sci.* 11, 1880-1889 (2018)), which is not a very new concept. Therefore, from both novelty and device performance perspectives, I think the paper does not reach the high standard set by Nature Communications. It may more suitable for other sister journals if the following concerns are addressed.

Response: We appreciate the reviewer's comments. The mismatch factor of solar simulator in our lab is ~ 1.017 , which guarantees reliability of the measured photocurrent. Also, all the integrated current densities from EQE spectra agree well with the J_{sc} values obtained from J - V curves within 5% deviation, which is consistent to that reported by Sang Il Seok et al. (*Science* 2017, 356, 1376) and Henry J. Snaith et al. (*Nat. Energy* 2017, 2, 17135) etc. Moreover, we invited Enli-tech company to re-calibrate both solar simulator and EQE equipment, the resulted current densities are fully consistent with those before. Thus, the photocurrent and EQE spectra in our manuscript are reasonable and accurate.

For the *Energy Environ. Sci.* 11, 1880-1889 (2018) reviewer mentioned, it reports commercial para-substituted benzenethiol molecules treated perovskite/Spiro-OMeTAD hole-transporting layer (HTL) interface. Differently, we focus on modification of perovskite/PCBM electron-transporting layer (ETL) interface by designing iodide ionized fullerene electrolyte (PCBB-3N-3I). More importantly, **cross-scale characterizations** were performed to understand interfacial engineering effect of PCBB-3N-3I on device performance and stability:

At molecule scale. DFT calculation was employed to optimize molecule geometry and adsorption energy of PCBB-3N-3I on charged defects of perovskite, which is used to illustrate assembly of PCBB-3N-3I with preferred orientation.

At nanoscale. Cross-sectional SKPM was performed to visualize effect of assembled PCBB-3N-3I generated dipole interlayer on reconfiguration of energy band alignment in *operando* devices.

At mesoscale. XPS was used to provide evidence of interaction between PCBB-3N-3I and Pb of perovskite; SFG spectra and UPS have been employed to probe the molecule assembly of PCBB-3N-3I and the resulted interfacial dipole.

At macroscale. Trap density, trap-assisted recombination and charge collection etc. in PCBB-3N-3I devices were investigated by several semiconductor analysis techniques.

All these results from cross-scale characterization techniques agree with each other. The microscopic chemical structure and molecular orientation at the interface are directly correlated with macroscopic device performance via nano- and mesoscopic interface dipole and energy band structure measurements. We believe that *in-operando* and *in-situ* cross-scale characterizations will promote understanding the importance of interfacial effects on device performance.

In response to reviewer's critical comments, we have cited the mentioned reference (*Energy Environ. Sci.* 2018, 11, 1880-1889) as ref 35 (in red font color) and made a discussion to emphasize the importance of our work in page 20 as follows:

“This work also reveals the critical role of cross-scale characterizations in device interface research. The microscopic chemical structure and molecular orientation at the interface are directly correlated with macroscopic device performance via nano- and mesoscopic interface dipole and energy band structure measurements. We believe similar *in-operando* and *in-situ* cross-scale characterizations will be tremendously beneficial to understanding and exploring interfacial effects in future thin-film electronic and optoelectronic devices.”

1. The authors claimed that perovskite/PCBB-3N-3I interface showed much stronger charge collection ability than that of the perovskite/PCBB-3N and perovskite/PCBM interface due to the large dipole moment. I wonder the energy levels of PCBB-3N-3I or PCBB-3N itself also play a critical role on the interfacial charge collection efficiency in the perovskite solar cells. Thus, the authors should give more information about the HOMO and LUMO energy of the PCBB-3N-3I and PCBB-3N molecules and carefully verify the reliability of the experimental results.

Response: We agree the reviewer's comment that HOMO and LUMO of PCBB-3N-3I/PCBB-3N may also play an important role in charge collection. **Figure Fa-b** are UPS spectra and tauc-plot of PCBB-3N-3I and PCBB-3N. Based on the extracted energy levels in Figure Fc, it seems slight energy barrier for electron collection with both PCBB-3N-3I and PCBB-3N. However, the ultrathin (~ 1 nm) assembled PCBB-3N-3I formed larger dipole at perovskite/PCBM interface could induce a downshift in

LUMO as evidenced from the UPS spectra in Figure Fd and cross-sectional SKPM in **Figure 3**. As seen from the interfacial energy band structure in Figure Fe, the PCBB-3N-3I would facilitate electron collection. In contrast, the random PCBB-3N with negligible dipole would deteriorate charge collection. All these results are consisted to the device performance.

We have added Figure F as Supplementary Fig. 17 and the discussion in page 18 as follows: “The larger downshift in E_{vac} with the assembled PCBB-3N-3I suggests formation of a dipole interlayer with negative end pointed towards perovskite and positive end pointed outside as proposed in Fig. 2c, which brings significant change in interfacial energy band structure, leading to a downshift in LUMO of PCBB-3N-3I to lower charge collection barrier.”

Figure F. **a** UPS spectra of PCBB-3N-3I and PCBB-3N. **b** Tauc-plot of MAPbI₃, PCBM, PCBB-3N-3I and PCBB-3N. **c** Energy level diagram of MAPbI₃, PCBB-3N-3I and PCBB-3N. **d** UPS spectra of MAPbI₃ before and after ultrathin (~ 1 nm) PCBM, PCBB-3N-3I and PCBB-3N treatment. **e** Interfacial energy band structure between MAPbI₃ and PCBM/PCBB-3N-3I/PCBB-3N.

2. Line 125: a typo of “standard derivations” for “standard deviations”.

Response: We are sorry for the mistake. We have corrected it (in red font color).

3. Fig 1d. shows that the EQE is increased differently according to the wavelength considered. The higher increase at high wavelength might indicate better charge

extraction and reduced recombination at the back interface with PCBM which is coherent with the phenomena reported throughout the article. However, it does not appear obvious to me why the low wavelength show a higher increase in EQE, because unchanged electric field in fig. 3f suggests that the front charge extraction at the interface with PTAA is not expected to be significantly impacted. Therefore, this higher EQE increase could indicate that there is after all an effect induced on the perovskite/PTAA interface or maybe that the optical absorption properties have been modified, which might be worth discussing.

Response: Thanks for the reviewer's comments. EQE is determined by three basic processes including light absorption, charge carrier separation and collection. According to reviewer's suggestion, absorption spectra of perovskite/PCBM before and after PCBB-3N-3I or PCBB-3N treatment was measured (**Figure G**), which shows little difference due to ultrathin PCBB-3N-3I or PCBB-3N (~ 1 nm) and unchanged crystallinity of perovskite. As for the long-wavelength light, low absorption coefficient of perovskite enables deep light penetration, which indicates that charge carriers would generate throughout the perovskite active layer. After incorporating PCBB-3N-3I dipole interlayer, the enhanced band bending and electric field at perovskite/PCBM interface could promote charge carrier separation and collection, thus resulting in significant improvement in EQE at the long-wavelength region. While for the short-wavelength light, relatively high absorption coefficient of perovskite leads to shallow light penetration and most of charge carriers generate near the PTAA/perovskite interface. Since band bending at PTAA/perovskite interface shows negligible change before and after PCBB-3N-3I or PCBB-3N treatment as evidenced by the cross-sectional SKPM, the charge carrier separation was little influenced. Therefore, the increased EQE at short-wavelength region with PCBB-3N-3I dipole interlayer could be explained by the improved charge carrier collection due to enhanced band bending and electric field at PCBM/perovskite interface. The improved EQE at short-wavelength region by modification of the back interface has also been reported by Shashank Priya et al. *Nano Lett.* 2019, 19, 3313–3320; Jinsong Huang et al. *Adv. Mater.* 2017, 29, 1604545.

We have added corresponding discussion in page 8 and 17 to explain the EQE improvement with PCBB-3N-3I and Figure G as Supplementary Fig. 3.

Page 8: “Figure 1d is the external quantum efficiency (EQE) spectra of the champion devices, which shows significant improvement with PCBB-3N-3I and severe deterioration with PCBB-3N.”

Page 17: “The enlarged band bending of perovskite in the PCBB-3N-3I device suggested a higher V_{bi} than that of the control device, which is consistent with the Mott-Schottky measured values, thus contributing to higher V_{oc} , J_{sc} and EQE.”

Figure G. Absorption spectra of MAPbI₃ with and w/o PCBB-3N-3I/PCBB-3N.

Reviewer #3 (Remarks to the Author):

The authors showed that the charged surface defects can be benign after passivation and further exploited to provide favorable interfacial energy band alignment leading to enhanced PEC in perovskite solar cell. Overall the discussion is good. The SKPM and UPS techniques used to show an “experimental” band diagram can be **valuable to the Solid State community as a whole**. Before the paper can be accepted, the authors should address the following questions:

1. Table 1: Did the authors measure or determine the device series and shunt resistances? If adding this to the table, it could provide more insight about device performance. For instance, I noticed a slightly shallower I - V curve for the device with the PCBB-3N capping layer compared to the control group (no capping layer), suggesting that the series resistance is higher when using PCBB-3N. However, the series resistance appears lower for the PCBB-3N-3I capping layer compared to the control group. This was explained later by the SKPM measurements, but the series resistance argument could corroborate this.

Response: We thank reviewer for the critical comment. By fitting the J - V curves of champion devices in Fig. 1b with equivalent circuit model, the series resistance and shunt resistance are $1.11 \Omega \text{ cm}^2$ and $1078 \Omega \text{ cm}^2$ for PCBM device, $0.48 \Omega \text{ cm}^2$ and $2241 \Omega \text{ cm}^2$ for PCBB-3N-3I device, $1.41 \Omega \text{ cm}^2$ and $982 \Omega \text{ cm}^2$ for PCBB-3N device, respectively, which also provides evidence to understand effect of PCBB-3N-3I/PCBB-3N treatment.

In response to the reviewer’s critical comment, we added the series resistance and shunt resistance into Table 1. We have also added discussion in page 7-8 as follows: “The change in device performance with PCBB-3N-3I/PCBB-3N is also evidenced by extracted series resistance and shunt resistance as seen in Table 1. The series resistance is smaller in PCBB-3N-3I device and larger in PCBB-3N device than that in control device.”

2. Line 122: “...at a scan rate of 0.02 V s^{-1} ...” Is there a reason for choosing this scan rate? Does scan rate have any impact on the device performance, as previous literature mentioned?

Response: We appreciate reviewer’s comment. As seen in **Figure H**, it shows little difference in J - V curves of all kinds of device at various scan rates of 0.005 V s^{-1} , 0.02 V s^{-1} , 0.2 V s^{-1} , 2.0 V s^{-1} . In our manuscript, a relative slow scan rate of 0.02 V s^{-1} was selected to exclude the interference of ion migration induced hysteresis.

We have added Figure H in Supplementary Figure 4 and corresponding discussion in page 7 as follow: “No significant hysteresis was seen in all the devices at different scan rates (Supplementary Fig. 4).”

Figure H. *J-V* curves of **a** control device, **b** PCBB-3N-3I device and **c** PCBB-3N device under AM 1.5G illumination at a scan rate of 0.005, 0.02, 0.2, 2.0 V s⁻¹ for both forward (0 - 1.2 V) and reverse scanning (1.2 - 0 V).

3. Line 235: “The assembly of PCBB-3N-3I forms ... and positive end pointed outside (black arrow)...” Is there any evidence that the positive end points outward, rather than towards a negative defect (i.e. PbI₃)? I’m not fully convinced on their claims of specific orientation of the fullerene component based on the data provided.

Response: We appreciate the reviewer’s critical comment on molecule orientation. We agree that some PCBB-3N-3I molecules may exhibit other molecular orientations that are not in the arrangement we described, because of the complicated surface defects and serious morphology fluctuation of perovskite film. Interestingly, most PCBB-3N-3I molecules exhibit preferred orientation as proposed in Fig. 2c based on DFT calculation, XPS and SFG spectra, which is evidenced by a downshift in vacuum level after PCBB-3N-3I treatment measured by both UPS and cross-sectional SKPM. The downshift in E_{vac} suggests the assembled PCBB-3N-3I forms a dipole interlayer with negative end pointed towards perovskite and positive end pointed outside as proposed in Fig. 2c.

We have added corresponding discussion in page 12-13 and 18 for better clarification this molecular orientation.

Page 12-13: “indicating the preferred molecular orientation on perovskite with complicated surface defects and serious morphology fluctuation”

Page 18: “The downshift in E_{vac} suggests the assembled PCBB-3N-3I forms a dipole interlayer with negative end pointed towards perovskite and positive end pointed outside as proposed in Fig. 2c,”

4. Fig. 3d: The surface potential difference in the “bulk” region of the perovskite is nearly uniform, even in the short-circuit condition (SCC, top of the image). This suggests that most band bending occurs close to the interface. The schematic of Fig. 4 (derived from UPS) also shows this, which is good. This is surprising, for what should be an “intrinsic” material.

Response: We appreciate the reviewer’s critical comment. The recipe in our work will produce n-type doping in perovskite based on UPS measured energy level of perovskite as seen in **Figure Fc**. The n-type doping in perovskite will reduce depletion layer width, resulting in significant band bending at perovskite/HTL (or ETL) interface and little change in surface potential (SP) across perovskite.

We have added the discussion in page 15 and UPS spectra in Supplementary Fig. 17 as follows: “which was attributed to n-type doping in perovskite based on ultraviolet photoelectron spectroscopy (UPS) (Supplementary Fig. 17) that narrows depletion layer width.”

5. Line 275: “The SP changed slightly across the perovskite film but dramatically dropped due to the existence of two junctions, which agrees with the behavior of p-i-n structure devices”. But in a typical p-i-n a large drift field is maintained. This comment is related to the one above.

Response: We appreciate the reviewer's expertise in energy band structure. We are sorry for the misused p-i-n structure. As shown above in response to the **Comment 4**, devices with n-type perovskite active layer consists of p-n junction at PTAA/perovskite interface and n-n junction at perovskite/PCBM interface, which explain SP drop at both interfaces. Also, the n-type doping in perovskite gives rise to a narrower width of depletion layer, resulting in little change in SP across perovskite, i.e. without significant drift field in the perovskite active layer.

We have replaced all the "p-i-n structure" with "inverted structure" in the title and the main text for better clarification (in red font color).

6. Fig. S2: The significant presence of PbI_2 in the XRD plots is concerning. It is quite surprising that their device performance was as high as reported based on the PbI_2 impurity peak. The authors should provide some comment on this.

Response: We thank reviewer's comment. Actually, excess PbI_2 in perovskite films have been widely demonstrated to be an efficient way to achieve high-performance. For example, Jingbi You *et al.* (*Adv. Mater.* 2017, 29, 1703852) systematically investigate effect of various PbI_2 content on device efficiency, and found that excess PbI_2 could passivate surface or grain boundary of perovskite film and facilitate charge transportation. As a result, a high content PbI_2 , showing an even higher XRD diffraction intensity than that of (100) crystal plane of perovskite, delivered the best efficiency exceeding 21%. Similar conclusion was also demonstrated by Sang Il Seok *et al.* in *Adv. Energy Mater.* 2016, 6, 1502104. In addition, the perovskite fabrication method in our manuscript were also adopted by many other groups like Tao Wang (*J. Mater. Chem. A*, 2017, 5, 9402; *Adv. Funct. Mater.*, 2017, 27, 1702613) and Yinhua Zhou (*J. Mater. Chem. A*, 2017, 5, 17632.), where obvious PbI_2 diffraction peaks was observed and showed a comparable control device performance with us. This method was demonstrated to have a high reproducibility. In our previous work (*Adv. Energy Mater.* 2018, 8, 1703054), the control device fabricated by the same method showing a comparable power conversion efficiency.

We have added corresponding discussion on the importance of small amount of excess PbI_2 in page 6 as follows: "Here, a small amount of excess PbI_2 was adopted, which could passivate the perovskite film and facilitate charge transport, thus resulting in superior device performance and mitigated hysteresis."³⁶

7. Line 370: "...can reduce trap-assisted recombination...". What are the direct evidence of the reduction of trap states?

8. Fig. S5: The author's method of Mott-Gurney analysis to measure the trap density in perovskite using their structures is questionable. It should be a metal-

semiconductor-metal structure, unless they can argue that the effects of transport layers can be negated. If so, they should remove their trap density argument as it is not well proven by their data.

Response: We appreciate the reviewer’s comments. Since both comments are related to trap density of states, we response them together here.

We agree that metal-semiconductor-metal structure is ideal case for Mott-Gurney analysis to extract trap density. However, it is difficult to get stable and reliable J - V curves in electron-only devices without charging transporting interlayers, which may be attributed to chemical reaction, poor interfacial contact and/or ion motion etc. Therefore, we have performed thermal admittance spectroscopy to extract trap density of states, which is effective on characterize defects of thin-film perovskite solar cells as reported by Henry Snaith et al. *Energy Environ. Sci.* 2016, 9, 490, Jingsong Huang et al. *Nat. Energy* 2017, 2, 17102 etc. **Figure I** shows that PCBB-3N-3I device exhibits lower trap density of states than that of control device and PCBB-3N device, indicating reduced surface defects of perovskite.

We have replaced SCLC data with Figure I as Supplementary Fig. 7 and revised discussion in page 9 as follows: “The difference in n of these devices was further corroborated by trap density of states extracted from thermal admittance spectroscopy (Supplementary Fig. 7). The lower trap density of states in PCBB-3N-3I device than that in control device and PCBB-3N device, indicating reduced surface defects of perovskite.”

Figure I. Trap density of states of control device, PCBB-3N-3I device and PCBB-3N device extracted from thermal admittance spectroscopy.

REVIEWERS' COMMENTS:

Reviewer #1 (Remarks to the Author):

The authors have performed additional studies to prove their concepts, according to the reviewer's comments.

Additional FT-IR, Raman, XPS, SFG results made the work more comprehensively elucidated and clear. Therefore, I think this work has decent value to interest readers in this field, and therefore agree to be published in Nature Communications.

However, recent references which are related to the development of interfacial layers for high-performance inverted structure PSCs should be added before publication.

Reviewer #2 (Remarks to the Author):

The author's response to the review comment is quite detailed and thorough. Regarding the device performance characterization like I-V curve and EQE curve, the author rechecked and re-calibrated the solar simulator and EQE equipment, which verified that the tested results were convincing and reasonable. Besides, the author put forward that the bright point of their work was the application of cross-scale characterization methods such as SKPM, XPS, SFG, UPS and TAS etc. The all-round characterization means explain the internal mechanism of the reconfiguration of the interfacial energy band structure. It is pointed out that the energy level was slightly modified with the dipole formed at the interface, facilitating charge collection. In addition, the author offered adequate evidence to support the EQE result.

However, I still have some doubts about the device performance data. In my opinion, the performance data would not be so high as shown in this work if the author used a shadow mask. Whether the data is consistent with that obtained with a mask remains to be verified.

Reviewer #3 (Remarks to the Author):

The revision looks very good, and answered most of the questions I raised.

It will be helpful that the authors could briefly comment on these two questions:

- Fig. S2: The significant presence of PbI₂ in the XRD plots is concerning. Most of our Perovskite layers do not show a significant presence of PbI₂. It is quite surprising (and somewhat concerning) that their device performance was as high as they reported based on the PbI₂ impurity peak.
- Fig. S5: Their method of Mott-Gurney analysis to measure the trap density in Perovskite using their structures may not be correct. It should be a metal-semiconductor-metal structure, unless they can argue that the effects of transport layers can be negated. They may want to remove their trap density argument as it is not well proven by their data. This figure is OK in the SI, because the main argument of trap density is shown by their capacitance-frequency plots in the SI. I'd suggest a slight re-wording or 1 – 2 sentence explanation in the S.I. as to how the figure is used mainly to corroborate the capacitance-frequency density of trap states plot.

Reviewers' comments:

Reviewer #1 (Remarks to the Author):

The authors have performed additional studies to prove their concepts, according to the reviewer's comments. Additional FT-IR, Raman, XPS, SFG results made the work more comprehensively elucidated and clear. Therefore, I think **this work have decent value to interest readers in this field, and therefore agree to be published in Nature Communications**. However, recent references which are related to the development of interfacial layers for high-performance inverted structure PSCs should be added before publication.

Response: We appreciate reviewer's positive comments. We have added interfacial layers related recent references as ref 28-29 based on reviewer's suggestion as follows: “²⁸. Wang Y. et al. Stabilizing heterostructures of soft perovskite semiconductors. *Science* **365**, 687-691 (2019). ²⁹. Liu T. et al. Tailoring vertical phase distribution of quasi-twodimensional perovskite films via surface modification of hole-transporting layer. *Nat. Commun.* 10, 878 (2019).”

Reviewer #2 (Remarks to the Author):

The author's response to the review comment is quite detailed and thorough. Regarding the device performance characterization like I-V curve and EQE curve, the author rechecked and re-calibrated the solar simulator and EQE equipment, which verified that the tested results were convincing and reasonable. Besides, the author put forward that the bright point of their work was the application of cross-scale characterization methods such as SKPM, XPS, SFG, UPS and TAS etc. The all-round characterization means explain the internal mechanism of the reconfiguration of the interfacial energy band structure. It is pointed out that the energy level was slightly modified with the dipole formed at the interface, facilitating charge collection. In addition, the author offered adequate evidence to support the EQE result.

However, I still have some doubts about the device performance data. In my opinion, the performance data would not be so high as shown in this work if the author used a shadow mask. Whether the data is consistent with that obtained with a mask remains to be verified.

Response: We thank reviewer's comments. Based on reviewer's suggestion, we measured J - V curves of devices with 0.049 cm^2 mask or w/o mask as seen in **Figure A**. The device performance measured with the mask show little difference that that obtained w/o mask (Table A), which corroborates the reliability of measurements.

We have added Figure A, Table A as Supplementary Fig. 5, Supplementary Table 4. We have added description in page 7 "The device performance with a mask was also investigated, which showed little difference than that w/o a mask and corroborated the reliability of measurements (Supplementary Fig. 5 and Supplementary Table 4)." and in Methods section in page 19 "To corroborate the reliability of device measurements, non-reflective mask with area of 0.049 cm^2 was also used to define the cell area."

Figure A. J - V curves of control device, PCBB-3N device and PCBB-3N-3I device under AM 1.5G illumination with 0.049 cm^2 mask or w/o mask.

Table A. Photovoltaic parameters of control device, PCBB-3N-3I device and PCBB-3N device with 0.049 cm² mask or w/o mask

	Mask	V_{oc} (V)	J_{sc} (mA cm ⁻²)	FF (%)	PCE (%)
control	w/o mask	1.067	22.38	73.33	17.51
	with mask	1.067	22.29	73.79	17.55
PCBB-3N-3I	w/o mask	1.105	23.29	80.86	20.81
	with mask	1.103	23.14	81.45	20.79
PCBB-3N	w/o mask	1.042	20.90	70.41	15.33
	with mask	1.039	20.73	71.06	15.30

Reviewer #3 (Remarks to the Author):

The revision looks very good, and answered most of the questions I raised. It will be helpful that the authors could briefly comment on these two questions:

Response: We thank reviewer's constructive comments to help us improve the manuscript. Based on reviewer's suggestion, we have added comments in the manuscript and Supplementary Information detailed as follows:

1. Fig. S2: The significant presence of PbI_2 in the XRD plots is concerning. Most of our Perovskite layers do not show a significant presence of PbI_2 . It is quite surprising (and somewhat concerning) that their device performance was as high as they reported based on the PbI_2 impurity peak.

Response: We have added ref 38 and discussion on the presence of PbI_2 in the XRD spectra in page 6 as follows: "Here, a small amount of excess PbI_2 was adopted, which gave rise to obvious PbI_2 peak in the XRD spectra of perovskite film (Supplementary Fig. 1).³⁸ The excess PbI_2 could passivate the perovskite film and facilitate charge transportation, thus resulting in superior device performance and mitigated hysteresis."³⁹

2. Fig. S5: Their method of Mott-Gurney analysis to measure the trap density in Perovskite using their structures may not be correct. It should be a metal-semiconductor-metal structure, unless they can argue that the effects of transport layers can be negated. They may want to remove their trap density argument as it is not well proven by their data. This figure is OK in the SI, because the main argument of trap density is shown by their capacitance-frequency plots in the SI. I'd suggest a slight re-wording or 1 – 2 sentence explanation in the S.I. as to how the figure is used mainly to corroborate the capacitance-frequency density of trap states plot.

Response: We have recovered trap density measurement based on Mott-Gurney analysis as **Supplementary Figure 8b** and added explanation in figure caption to corroborate the capacitance-frequency density of trap states plot as follows:

Supplementary Figure 8. Trap density measurements. (a) Trap density of states (tDoS) of control device, PCBB-3N-3I device and PCBB-3N device extracted from thermal admittance spectroscopy. (b) J - V curves of electron-only devices with a structure of ITO/TiO_x/MAPbI₃/PCBM/Al with and w/o PCBB-3N-3I/PCBB-3N treatment. The smaller trap-filled limited voltage (V_{TFL}) with PCBB-3N-3I treatment indicates reduced trap density n_t , which agrees with lower tDOS in PCBB-3N-3I device. The little change in V_{TFL} with PCBB-3N treatment means comparable n_t , which supports little difference in tDoS between PCBB-3N-3I device and control device.